# Roughness Evaluation of Burnished Topography with a Precise Definition of the S-L Surface

Przemysław Podulka 

Faculty of Mechanical Engineering and Aeronautics, Rzeszow University of Technology, Powstancow Warszawy 8 Street, 35-959 Rzeszów, Poland; p.podulka@prz.edu.pl; Tel.: +48-17-743-2537

**Abstract:** Studies of surface topography including processes of measurement and data analysis have an influence on the description of machined parts with their tribological performance. Usually, surface roughness is analysed when a scale-limited (S-L) surface, excluding short (S-) and length (L-) components from the raw measured data, is defined. Errors in the precise definition of the S-L surface can cause the false estimation of detail properties, especially its tribological performance. Errors can arise when the surface contains some burnished details such as oil pockets, dimples, scratches, or, generally, deep or wide features. The validation of proposed methods for S-L surface definition can also affect the accuracy of the ISO 25178 surface topography parameter calculation. It was found that the application of commonly used procedures, available in commercial software (e.g., least-square fitted cylinder element or polynomial planes, regular or robust Gaussian regression, spline, median or fast Fourier transform filters) can be suitable for precise S-L surface definition. However, some additional analyses, based on power spectral densities, autocorrelation function, texture direction graphs, or spectral characterisation, are strongly required. The effect of the definition of the S-L surface on the values of the ISO 25178 parameters was also comprehensively studied. Some proposals of guidance on how to define an appropriate S-L surface with, respectively, an objective evaluation of surface roughness parameters, were also presented.

**Keywords:** surface topography; surface texture; burnishing; oil pockets; dimples; measurement; measurement error; measurement noise; S-filter; L-filter; S-L surface

## 1. Introduction

The analysis of surface topography can be valuable in the characterisation of the tribological performance of machined surfaces. Valuable information including those when the surface is generated can be received straight from the surface topography characterisation such as wear resistance [1], lubricant retention [2], sealing [3], friction [4], energy consumption [5], eco-friendly applications and strategies [6], or functional performance [7] in general. The evaluation of surface topography can be significant when there is an assessment of additively manufactured parts [8,9]. Furthermore, in many cases, it can also be classified as a fingerprint of the manufacturing process [10].

When analysing the surface topography, there are many actions required, some of which can be validated separately. First, many errors can occur [11] when surface topography is measured, considering stylus [12] or non-contact [13] methods. Not only is precise measuring equipment required, but errors in the processing of the received raw measured data can cause a false estimation of the properties of the machined parts. From this perspective, accurate procedures for data characterisation should also be defined. Considering surfaces after different machining processes, it is difficult to define one general procedure for the raw measured data analysis. Regarding this matter, detailed guidance can play a significant role.

The whole process of surface topography analysis including, especially the characterisation of machined parts (surfaces), can be roughly divided into several, separate actions,

where mutual influence can occur. A necessary preliminary to the numerical assessment of surfaces or their selected profiles is to extract the frequency components, defined as the roughness and to respectively eliminate those that would be irrelevant such as form [14]. The surface topography of machined parts is usually analysed when a form including the shape and waviness is extracted [15]. However, the process of the removal of insignificant components of surface topography can be fraught with many errors, especially when the surface contains some dimples or, generally, deep and wide features [16]. The effect of the depth, size, or density of various features on the results of surface filtering with the general (available in the commercial software) methods has been widely studied previously [17].

It was found that reducing the errors in the raw measured data processing can be placed on an equal footing with errors in the manufacturing process, where an improper data processing approach can lead to the false estimation of surface properties, and the classification of properly made parts as lack and its rejection. When receiving the surface roughness parameters, an appropriate definition of the S-L surface [18], often described as a scale-limited surface, is required. The S-L surface is received when removing small-scale and large-scale lateral components from the primary surface by the application of S-operators (e.g., S-filter) and L-operators (L-filter), respectively. Both operations are considered separately.

The L-surface, when removing the long-scale components from the measured data, can be received by the application of digital filtering, often characterised by Gaussian filters [19]. Many Gaussian filter limitations were resolved when introducing its robust modifications [20,21], especially for surfaces containing some deep features. There were many digital filters provided for the extraction of form from the data such as spline [22,23], wavelet [24,25], morphological [26,27], fast Fourier transform (FFTF) [28,29], and many others [30]. Very popular in recent studies is multi-scale characterisation [31–33] or analysis based on feature consideration [34,35]. These techniques separate the received raw measured data in various frequencies using different methods, and filters in some cases [36].

Alternatively to filters, a form removal process can be provided with a defining reference plane [37]. This type of surface can be proposed by the application of the least-squares fitting methods [38]. When cylindrical details are measured, the reference plane can be obviously fitted by the cylindrical shape [39], polynomials with different degrees [40], or other planes suitable for representing the waviness [41,42]. The application of least-square fitting procedures can reduce the distortion of the deep or wide features, especially when they are located near (on) the edge of the analysed detail [43,44], compared to the regular digital filtering methods. However, guidance on how to use regular (e.g., those available in the commercial software) methods in the definition of a proper L-surface is required.

Second, the S-surface received when an S-operator (e.g., S-filter [45]) is applied must also consider that it has been found that the occurrence of S-components can radically influence the values of the ST parameters [46]. One of the components included in the small-scale lateral surface (S-surface) is the high-frequency measurement noise. Generally, according to the ISO standard [47], measurement noise can be defined as the noise added to the output signal occurring during the normal use of the measuring instrument. When creating a standard reference frame for describing and reducing the measurement noise, it is necessary to define it, along with the associated measurement bandwidth [48,49].

One of the often analysed bandwidths for measurement noise studies is in the high-frequency domain. Generally, the high-frequency measurement noise can be caused by the instability of the mechanics with any influences from the environment [50]. Nevertheless, in most cases, the high-frequency noise is caused by vibrations [51]. This problem seems to be even more significant when in situ measurement is provided [52,53]. Some solutions can be proposed with the isolation [54–56] or suppression [57] of vibration sources. However, it is difficult to provide a general procedure for the selection of the method for the S-surface definition.

One of the crucial tasks required to be resolved is to propose a method for a precise definition of the S-L surface for the textures containing some deep or wide features. It was observed in previous studies that the false estimation of the L-surface can enlarge the errors in the evaluation of the ISO 25178 surface topography parameters for various surfaces (e.g., turned [58], ground [59], laser-textured [60] and other topographies [61]). Plenty of detailed proposals can be presented with a thresholding method [62]. Generally, the threshold function can be used to segment the top and bottom surfaces (e.g., when the instrument calibration is provided [63]). This technique can be applied when a separation of the plateau and valley part of the laser-textured surfaces is proposed to also minimize the errors of the measurement errors [64].

In the whole process of calculating the surface roughness parameters, the accuracy of the ST evaluation increases when the precision in the S-L surface definition also increases. Reducing the errors in the S-L surface evaluation can be especially significant when functional properties are studied. The results of the calculation of surface topography parameters can be significantly influenced by the precision in the definition of the S-L surface. Both errors, those received when the S-surface and L-surface are defined, can be valuable in the dimple characterisation. There are many procedures for the elimination of the irrelevant components (like form, shape, waviness, and measurement noise) from the raw measured data, nevertheless, errors in the ISO 25178 parameter calculation were not studied in detail when the distortion of selected features (e.g., dimples) was considered. The effect of the type of surface, after various finishing treatments (e.g., plateau-honed, turned, or isotropic), containing additionally burnished oil pockets on the results of dimple distortion are presented in this paper. Some guidance on how to reduce the data distortion when defining the S-L surface is also proposed.

## 2. Materials and Methods

### 2.1. Analysed Details

Two types of surface finishing were analysed—plateau-honed, with a cross-*hatch* pattern equal to $60°$, turned, and isotropic details. Surfaces containing dimples of different sizes were considered. The dimple's average width and average depth were between 0.2 and 0.8 mm and between 10 and 100 μm, respectively. The length and width of the dimples were equal and they were circular, so, correspondingly, their width was only considered. More than 20 measured surfaces were studied, but few of them are presented in detail.

In Figure 1, the contour map plots (a–c), isometric views (d–f), material ratio curves (g–i), autocorrelation function graphs (j–l), and ISO 25178 surface topography parameters (including Sk family) are presented for three various surfaces. The first surface, presented in the left column, was a plateau-honed cylinder liner with additionally created oil pockets received by the burnishing technique, where correspondingly, the dimple width and depth were, on average, around 0.3 mm and 5 μm. The second surface, located in the middle column, was an isotropic cylindrical surface with modelled dimples with a width of around 0.8 mm and a depth of 60 μm. The last and third surface was a turned cylindrical surface containing additionally burnished dimples with a 0.7 mm and 100 μm width and depth, respectively. The area density of the oil pockets was smaller than 20% (this has been studied in many previous papers considering tribological performance) for all of the surfaces studied.

The values of the following ISO 25178 surface topography parameters were measured, calculated, and considered: arithmetic mean height $Sa$, auto-correlation length $Sal$, surface bearing index $Sbi$, core fluid retention index $Sci$, root mean square gradient $Sdq$, developed interfacial areal ratio $Sdr$, core roughness depth $Sk$, kurtosis $Sku$, maximum peak height $Sp$, arithmetic mean peak curvature $Spc$, peak density $Spd$, reduced summit height $Spk$, root mean square height $Sq$, upper material ratio $Sr1$, lower material ratio $Sr2$, skewness $Ssk$, texture direction $Std$, texture parameter $Str$, maximum valley depth $Sv$, valley fluid retention index $Svi$, reduced valley depth $Svk$, and the maximum height of surface $Sz$.

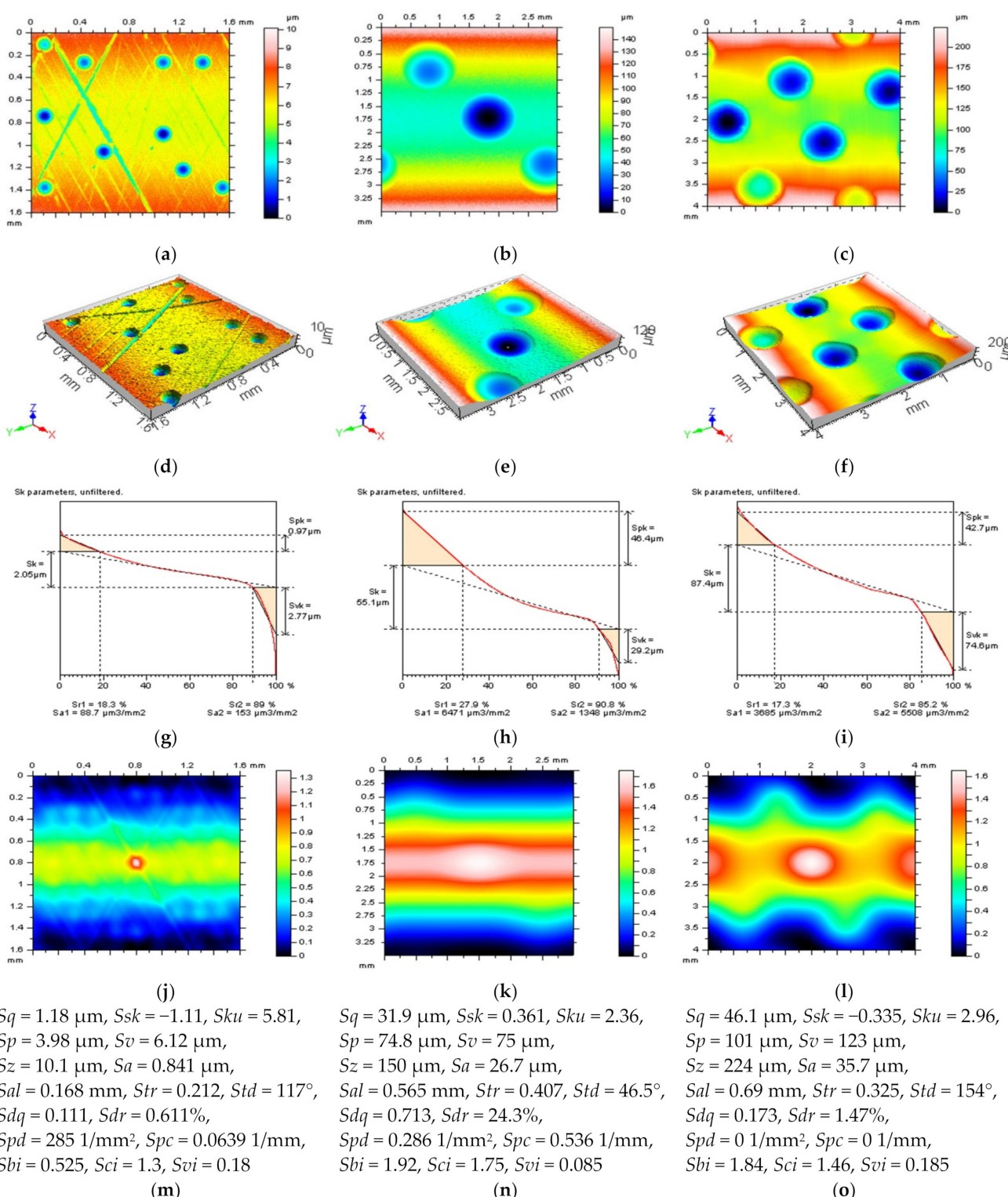

**Figure 1.** Contour map plots (**a**–**c**), isometric views (**d**–**f**), material ratio curves (**g**–**i**), 3D ACF (**j**–**l**) and the ISO 25178 surface topography parameters (**m**–**o**), received from a cylindrical surface with additionally burnished dimples with different sizes and densities.

## 2.2. Measurement Process

The analysed details were measured by stylus and optical techniques. The stylus instrument was a Talyscan 150 containing a nominal tip radius of 2 µm, and a height resolution of 10 nm. The measured area was 5 by 5 mm with $1000 \times 1000$ measured points. For the stylus measurement, the sampling interval was equal to 5 µm. The measurement speed was 0.75 mm/s. The effect of the measurement velocity was not studied in this research, and has been analysed in previous papers [10], so it was not included in the subject of the presented research.

The non-contact measurement device was a white light interferometer, Talysurf CCI Lite. Its height resolution was 0.01 nm and the measured area was 3.35 by 3.35 mm. The $1024 \times 1024$ measured points were received. The spacing was 3.27 µm. In this study, the effect of both sampling and spacing on the values of 3D surface roughness parameters was not analysed in the research presented.

Furthermore, all the details (surfaces) were carefully analysed to detect the individual peak (spike) errors from the raw measured data. This type of error was observed for the optical measurement technique. These were removed by the thresholding method with a material ratio from 0.13% to 99.87%, as proposed in the previous analysis [65].

## 2.3. Introduction to the Applied Methods

For the areal form removal including the extraction of the shape and waviness, various, available in commercial software, methods were applied. For the cylindrical surfaces (e.g., plateau-honed cylinder liner details), fitting the cylinder element seems to be an obvious solution [39]. In the considered examples, the least-square fitted cylindrical (LSFC) method was applied. Very popular in the areal form removal of machined parts are polynomials [40]. The author of [66] proposed the use of a polynomial of second degree (POLY2) for the areal form removal of curved surfaces. The greater degrees of polynomial planes (e.g., fourth (POLY4), and sixth (POLY6)) could distort oil pockets and, respectively, would cause the false estimation of the surface roughness parameters, and, unfortunately, the classification of properly made parts as lack and rejection. However, the ex-aggregation of wholes was reduced when their sizes were smaller. For this reason, analysis of the application of the polynomial method from the second to sixth degrees seems to be a suitable solution.

In contrast to the least-square methods, surface filtering has been commonly proposed in many previous studies [43]. Gaussian (GRF) and robust Gaussian (RGRF) regression filters have been widely described [19–21] and their suitability for surface roughness evaluation has been examined in both past [67,68] and present [69] studies. For surface roughness characterisation, the spline filters can also be highly advantageous [70].

For the validation of the methods for surface filtering and, respectively, defining both the S-surface and L-Surface, some general, available in commercial software functions have been proposed. Directly applicable and often used for the characterisation of surface roughness is an autocorrelation function (ACF). The analysis of surface roughness can be presented with the help of the evaluation of their autocorrelation function, where methods of deducing higher orders of autocorrelation lengths can be valuable to evaluate the non-random distribution of roughness amplitudes [71]. Connecting the ACF with specified standard deviation, skewness, and kurtosis, based on the digital FIR (finite impulse response) technique (filter), can be applied to non-Gaussian surface generation to characterise the mixed lubrication [72]. Many related to the ACF methods such as the structure-function, without some of its disadvantages, have been proposed as a means to also quantify the variations in the surface texture [73]. However, it has mainly been found that the ACF, in both its 3D (areal) and 2D (profile) forms, can be exceedingly valuable in the characterisation of high-frequency measurement noise when detection and reduction are proposed [10], especially when its central part shape is considered [46]. It was found that, in some cases, both 2D and 3D analysis are important in surface characterisation [74].

Alternatively to ACF, the power spectral density (PSD) can provide a lot of relevant information on the analysed surfaces. PSD characterisation provides both lateral and vertical

signals captured from atomic force microscope (AFM) images [75], where AFM is one of the most popular techniques for metrology measurements with nanoscale resolution. Considering a draft international drawing standard for surface texture, the PSD has been designated as the preferred quantity for specifying surface roughness [76]. Moreover, the roughness, measured with different instruments and techniques, could be directly compared when two-dimensional PSD functions are calculated from the digitised measurement data with the simultaneously obtained RMS (root mean square) roughness by integrating areas under the PSD curves between the fixed upper and lower band limits [77]. A multi-spectrum analysis method can be used in the investigation of the formation of surface roughness in ultra-precision diamond turning [78]. Modification of the spectrum method [79], the singular spectrum analysis (SSA), can be used for enhanced surface roughness monitoring of vibration signals generated in workpiece-cutting tool interaction in CNC finish turning operations [80]. The surface roughness spectrum can be analysed by the fast Fourier transform (FFT) method to identify how different wavelengths of roughness behave [81]. However, the limitation of the traditional FFT spectrum method must be mentioned, that is, is to describe the frequency characteristics in the entire time domain, which is limited in many studies [82]. Some experimental results with PSD application have indicated that the profile of the tool marks is distorted by the effect of swelling of the materials being cut when the diamond turning of single crystals was considered [83]. The application of PSD makes AFM a powerful tool that provides quantitative information not only on the height deviation of the roughness profile, but the spatial extent of the height variations in the roughness profile [84]. Nevertheless, the PSD has been found to be an appropriate method to also improve the definition of high-frequency measurement errors from the results of surface topography measurements [34].

A comprehensive analysis of the surface roughness can be markedly improved through the application of the texture direction (TD) graphs. It was demonstrated that the method of surface roughness measurement, especially based on the overlap degree of the colour image, had relatively high accuracy and a relatively wide measurement range and can be robust to the brightness of the light source and to the texture direction [85]. A three-dimensional roughness measurement method based on mathematical morphology can ignore surface directionality, and as such, applies to directional and non-directional surfaces [86]. Generally, the texture direction, simultaneously represented by the Std parameter, also plays an important role in determining the frictional behaviour of the surfaces [87]. The validation of the method for areal form removal, defined as the L-surface, analysis of the isometric view of a surface, can be especially valuable but only for an experienced user. In Figure 2, the surface (left column) after areal form removal by application of regular methods is presented. From this analysis, the usage of the polynomial of the second degree seems to be the best solution as some form components were found (not entirely removed) for other methods. The distortion of dimples increased when digital filters (GRF, RGRF, SF) were applied to cylindrical surfaces containing deep and wide dimples (Figure 3). The ex-aggregation was especially visible when oil reservoirs were located near/on the edge of the analysed detail. Least-square fitting methods (LSFC and POLY2) could provide more suitable results, nevertheless, fitting of the cylindrical element (LSFC) caused that form to not be eliminated entirely (Figure 3a).

Reduction in errors in the definition of the S-surface could be received when analysing the noise surface (NS) properties, which were comprehensively studied and widely defined in [34]. In Figure 4, the S-L surfaces were obtained by the application of various methods. A properly defined L-surface (Figure 4d) was not valuable in the S-L surface description when an S-surface (Figure 4g) was calculated erroneously. On the other hand, the proper S-surface (Figure 4i) was useless in the S-L surface characterisation if the L-surface (Figure 4f) was falsely estimated. Both surfaces must be determined precisely.

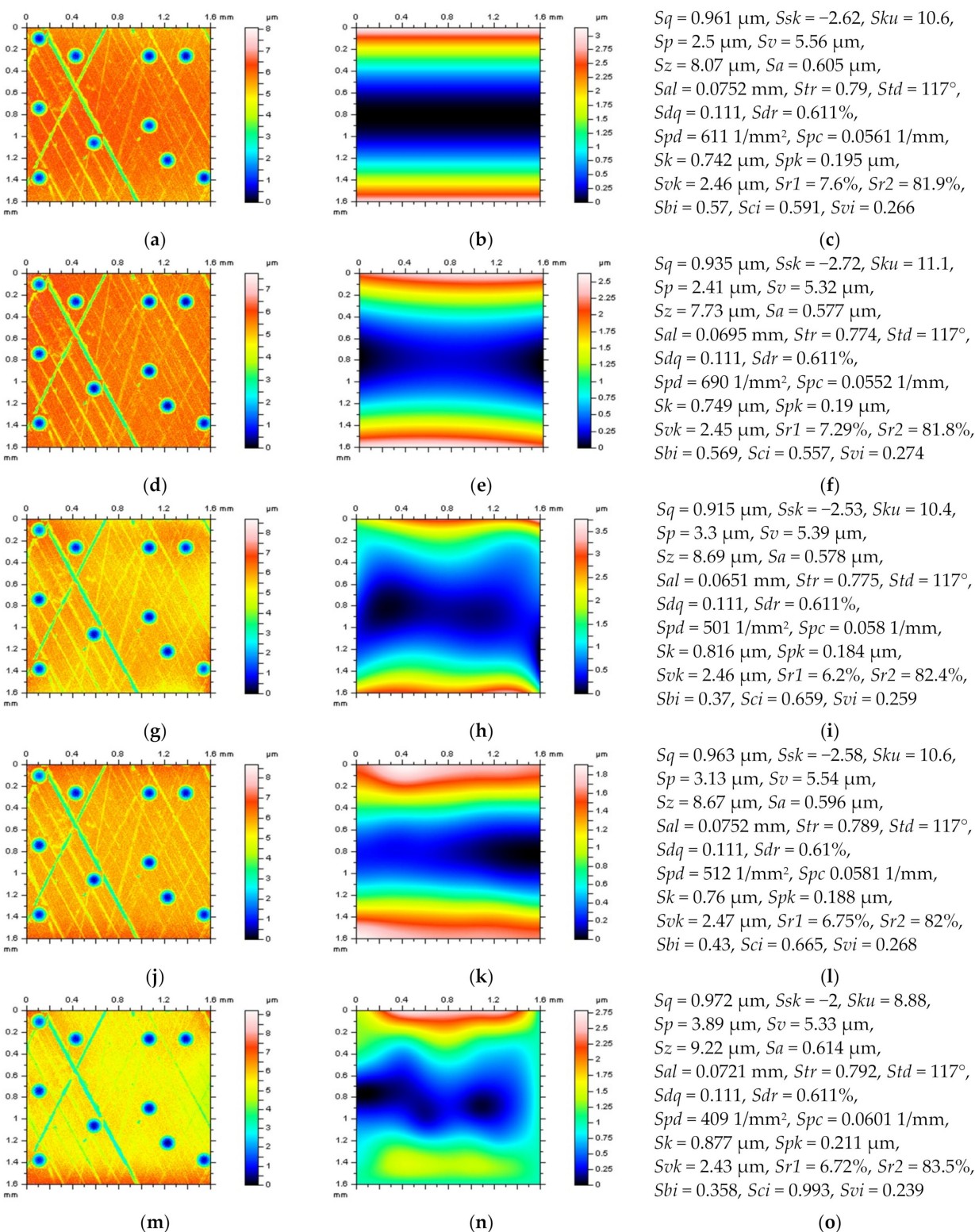

**Figure 2.** Contour map plots of the surface after the areal form removal (left column), reference planes (middle), and ISO 25178 parameters (right column), received from a plateau-honed cylinder liner surface with oil pockets created by burnishing techniques, after the application of LSFC (**a**–**c**), POLY2 (**d**–**f**), POLY6 (**g**–**i**), RGRF (**j**–**l**), and SF (**m**–**o**), cutoff = 0.8 mm.

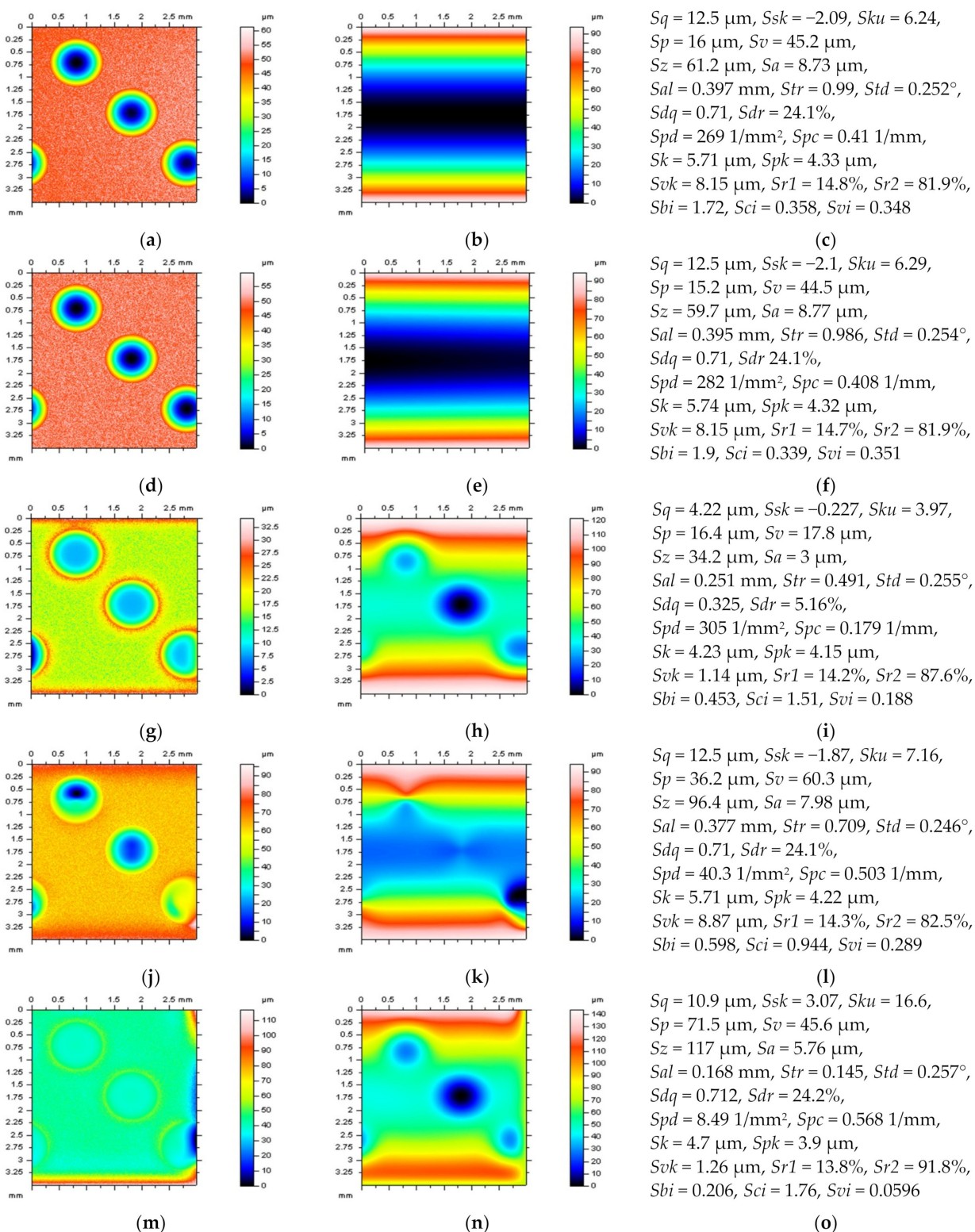

**Figure 3.** Contour map plots of the surface after the areal form removal (left column), reference planes (middle), and ISO 25178 parameters (right column), obtained from an isotropic cylindrical surface with wide and deep dimples, after the application of LSFC (**a**–**c**), POLY4 (**d**–**f**), GRF (**g**–**i**), RGRF (**j**–**l**), and SF (**m**–**o**), cutoff = 0.8 mm.

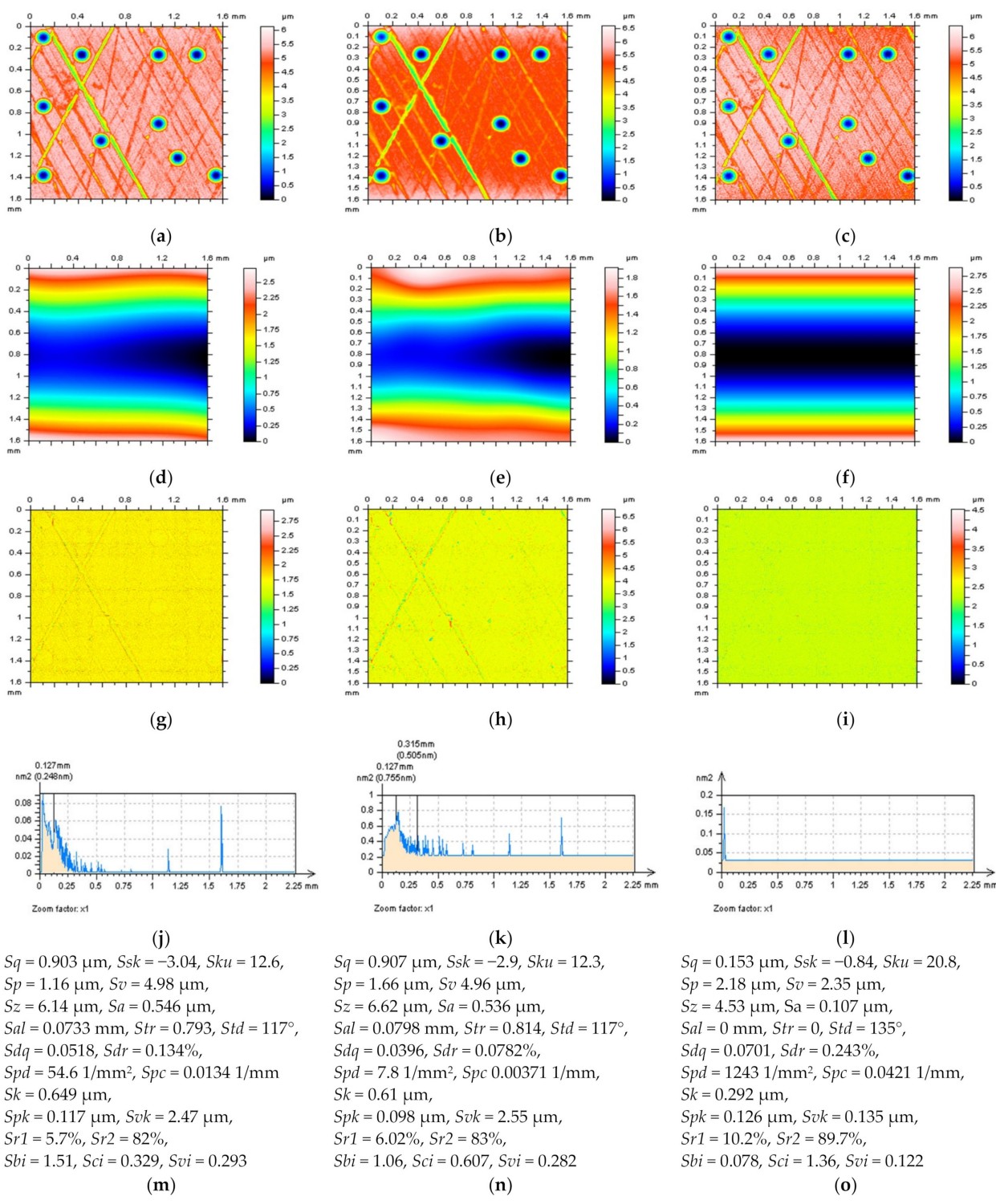

$Sq$ = 0.903 μm, $Ssk$ = −3.04, $Sku$ = 12.6,
$Sp$ = 1.16 μm, $Sv$ = 4.98 μm,
$Sz$ = 6.14 μm, $Sa$ = 0.546 μm,
$Sal$ = 0.0733 mm, $Str$ = 0.793, $Std$ = 117°,
$Sdq$ = 0.0518, $Sdr$ = 0.134%,
$Spd$ = 54.6 1/mm², $Spc$ = 0.0134 1/mm,
$Sk$ = 0.649 μm,
$Spk$ = 0.117 μm, $Svk$ = 2.47 μm,
$Sr1$ = 5.7%, $Sr2$ = 82%,
$Sbi$ = 1.51, $Sci$ = 0.329, $Svi$ = 0.293

(**m**)

$Sq$ = 0.907 μm, $Ssk$ = −2.9, $Sku$ = 12.3,
$Sp$ = 1.66 μm, $Sv$ = 4.96 μm,
$Sz$ = 6.62 μm, $Sa$ = 0.536 μm,
$Sal$ = 0.0798 mm, $Str$ = 0.814, $Std$ = 117°,
$Sdq$ = 0.0396, $Sdr$ = 0.0782%,
$Spd$ = 7.8 1/mm², $Spc$ 0.00371 1/mm,
$Sk$ = 0.61 μm,
$Spk$ = 0.098 μm, $Svk$ = 2.55 μm,
$Sr1$ = 6.02%, $Sr2$ = 83%,
$Sbi$ = 1.06, $Sci$ = 0.607, $Svi$ = 0.282

(**n**)

$Sq$ = 0.153 μm, $Ssk$ = −0.84, $Sku$ = 20.8,
$Sp$ = 2.18 μm, $Sv$ = 2.35 μm,
$Sz$ = 4.53 μm, $Sa$ = 0.107 μm,
$Sal$ = 0 mm, $Str$ = 0, $Std$ = 135°,
$Sdq$ = 0.0701, $Sdr$ = 0.243%,
$Spd$ = 1243 1/mm², $Spc$ = 0.0421 1/mm,
$Sk$ = 0.292 μm,
$Spk$ = 0.126 μm, $Svk$ = 0.135 μm,
$Sr1$ = 10.2%, $Sr2$ = 89.7%,
$Sbi$ = 0.078, $Sci$ = 1.36, $Svi$ = 0.122

(**o**)

**Figure 4.** S-L surface (**a**–**c**), removed form (**d**–**f**), NS (**g**–**i**), the PSD of NS (**j**–**l**), and ISO 25178 parameters of the S-L surface (**m**–**o**), received after application of POLY4 and GRF (cutoff = 0.025 mm) (left column), RGRF (L-operator with cutoff = 0.8 mm and S-operator with 0.025 mm) (middle column) and LSFC with FFTF (cutoff = 0.025 mm).

## 3. Results

Studies of the surface topography were divided into three main parts. First, the L-surface was defined with proposals for a method to reduce the effect of dimple distortion (Section 3.1). Second, the approach for the definition of the S-surface with minimisation of errors in the high-frequency measurement noise detection and removal is presented (Section 3.2). Finally, in the last subsection, the proposals for the reduction of errors in a definition of the S-L surface were raised with a profile characterisation (Section 3.3).

### 3.1. Definition of an Areal Form Removal Algorithm for a Precise L-Surface Definition

One of the main issues in the reduction of errors in the definition of an L-surface is to appropriately analyse the area where the features occur (e.g., oil pockets, dimples, oil reservoirs, scratches, or generally valleys). Plateau-honed cylinder liners are types of surfaces that contain a lot of these features. The degree of processed data ex-aggregation increases according to the enlargement in the size of each feature, however, this dependence is especially noticeable when deep/wide oil pockets/dimples are considered [17].

It was found in previous studies [16] that the false estimation of the reference plane for the surfaces containing burnished dimples could affect a selected group of parameters the most. First, growth in dimple distortion caused considerable differences in the amplitude (height) parameters, especially *Sp* and *Sv*, respectively, for the Sk group parameters, *Sk*, *Spk*, and *Svk*. Usually, the distortion of the dimples causes a huge exaggeration of parameters strictly related to the valleys. The values of the *Sv* and *Svk* parameters were distorted by about 100%. An application of digital filtering, whether containing the robust modification or not, caused a flatness of the oil pockets [16], which affected the decrease in the values of the *Sv* and *Svk* parameters. False calculation of these roughness parameters, correspondingly, had an impact on the Sk parameter. In practice, all of the Sk group parameters were received with distorted values [14].

In Figure 5, contour map plots of the turned cylindrical surface with additionally burnished deep and wide dimples are presented (left column) with ISO 25178 parameters (middle) and selected profiles (right column). Some of the errors in the definition of the L-surface can be visible from the analysis of the contour map plots of the surface. Regarding this, the application of all of the regular filters (e.g., GRF, RGRF, or SF) caused a serious distortion of dimples (Figure 5g,j,m). However, the location of these features influenced the degree of the deformation. Usually, the distortion increased if the dimple was positioned near/on the edge [88] of the considered detail. Moreover, for the Gaussian (GRF and RGRF) filters, the edge-effect had an enormous influence. Despite the location of the oil pockets, all of the applied filters did not allow us to receive a properly calculated reference plane.

In comparison with regular digital filtering methods, the least-square fitting schemes provided more persuasive results (Figure 5a,b). For each, the dimple exaggeration was substantially reduced. When a cylindrical shape (LSFC) was fitted, the distortion occurred in the areas near the edge, but only when more than one dimple was located nearby—this is indicated by the arrow. The most encouraging results were obtained when a polynomial reference plane (POLY2) was defined. However, both methods were supported by the valley-excluding method (VEM) [43]. Improvement in the reduction in dimple deformations was particularly visible in the profile exploration.

The validation of the methods applied for an L-surface calculation could be improved with an analysis of the ACF graph (Figure 6). It was observed that the surface after the application of a proper method for form removal should be flat [40]. The same property was observed for the 3D ACF defined for those surfaces (Figure 6m–o)—if a form was removed entirely, the shape of ACF was also close to the flat. However, this property was received when the ACFs were thresholded. These techniques, and especially their validity, have been widely introduced and considered for surface roughness analysis in many previous studies [35,64,89,90].

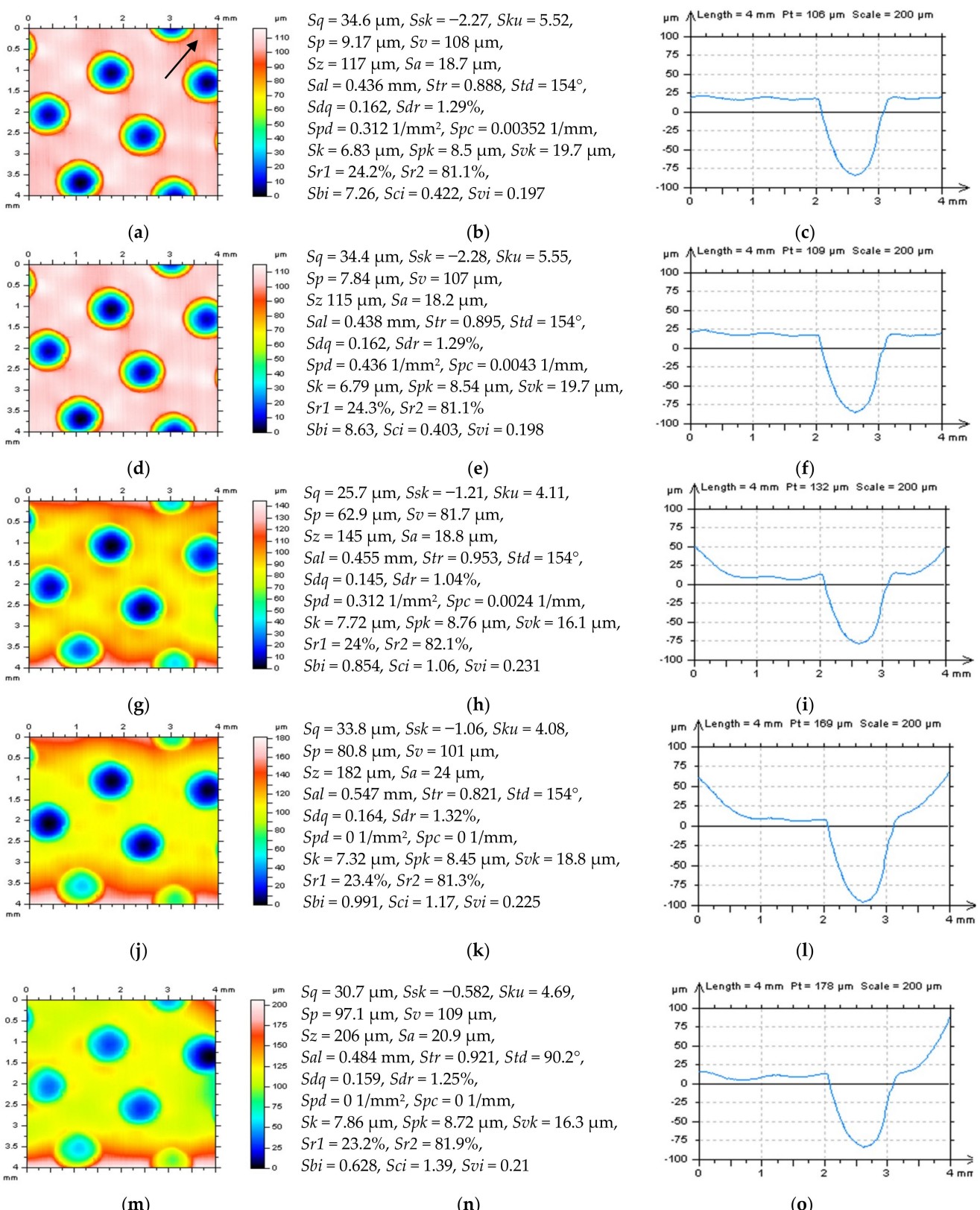

**Figure 5.** L-surfaces (left column), their ISO 25178 parameters (middle), and selected profiles (right column) received after the application of LSFC (**a**–**c**) and POLY2 (**d**–**f**) with VEM, GRF (**g**–**i**), RGRF (**j**–**l**), and SF (**m**–**o**) with a cutoff = 2.5 mm.

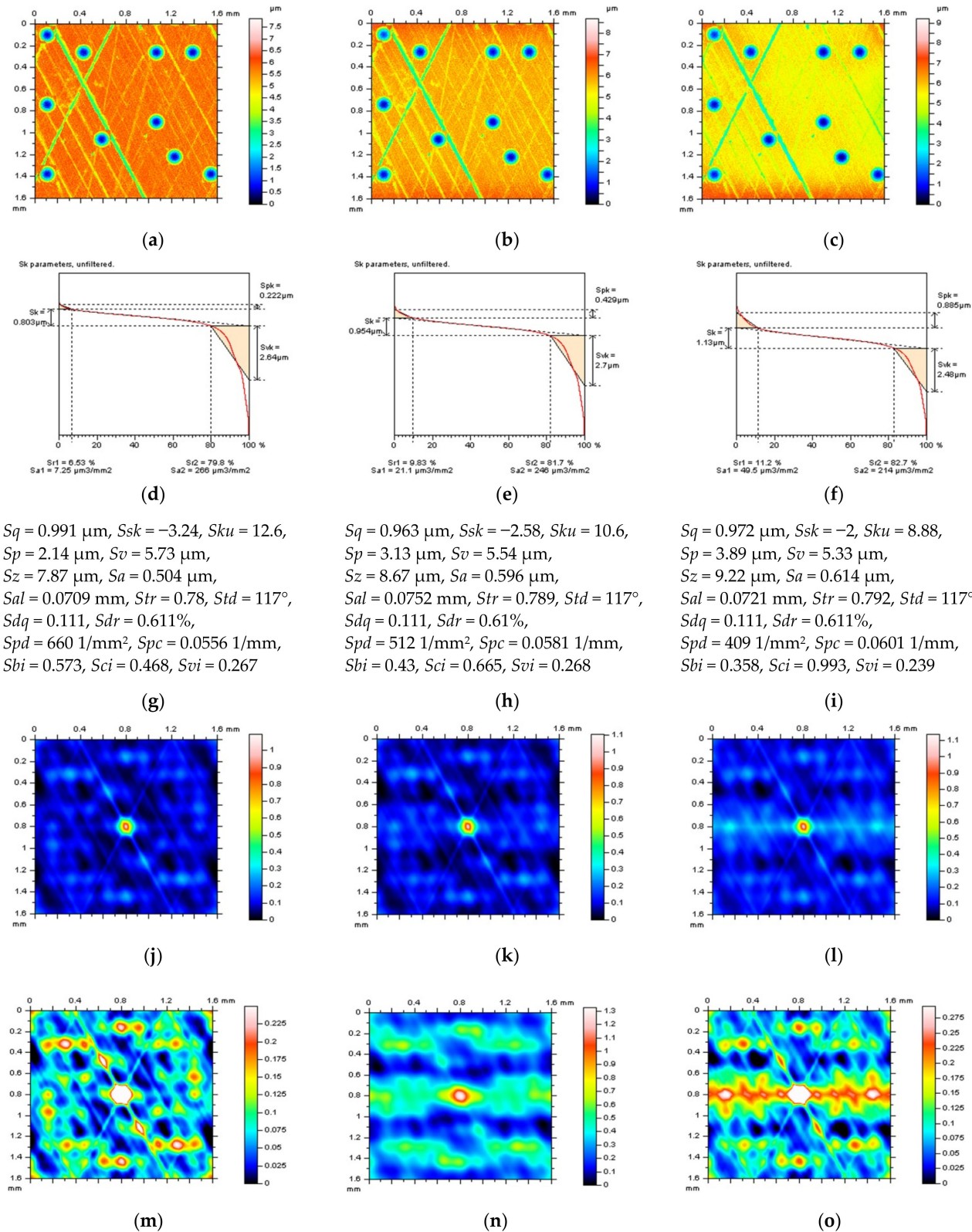

Sq = 0.991 μm, Ssk = −3.24, Sku = 12.6,
Sp = 2.14 μm, Sv = 5.73 μm,
Sz = 7.87 μm, Sa = 0.504 μm,
Sal = 0.0709 mm, Str = 0.78, Std = 117°,
Sdq = 0.111, Sdr = 0.611%,
Spd = 660 1/mm², Spc = 0.0556 1/mm,
Sbi = 0.573, Sci = 0.468, Svi = 0.267

(**g**)

Sq = 0.963 μm, Ssk = −2.58, Sku = 10.6,
Sp = 3.13 μm, Sv = 5.54 μm,
Sz = 8.67 μm, Sa = 0.596 μm,
Sal = 0.0752 mm, Str = 0.789, Std = 117°,
Sdq = 0.111, Sdr = 0.61%,
Spd = 512 1/mm², Spc = 0.0581 1/mm,
Sbi = 0.43, Sci = 0.665, Svi = 0.268

(**h**)

Sq = 0.972 μm, Ssk = −2, Sku = 8.88,
Sp = 3.89 μm, Sv = 5.33 μm,
Sz = 9.22 μm, Sa = 0.614 μm,
Sal = 0.0721 mm, Str = 0.792, Std = 117°,
Sdq = 0.111, Sdr = 0.611%,
Spd = 409 1/mm², Spc = 0.0601 1/mm,
Sbi = 0.358, Sci = 0.993, Svi = 0.239

(**i**)

**Figure 6.** L-surfaces (**a**–**c**), their material ratio curves (**d**–**f**), and ISO 25178 parameters (**g**–**i**), the 3D ACF of L-surfaces (**j**–**l**) and their thresholded (1–99%) parts (**m**–**o**), received after the areal form removal by application of POLY2 (left column), RGRF (middle column), and SF (right column), cutoff = 0.8 mm.

*3.2. Selection of Procedure for a High-Frequency Errors Reduction for S-Surface Definition*

The process of a high-frequency measurement noise reduction can be roughly divided into two separate but dependent subprocesses. First, the noise must be properly defined and then a suitable method can be applied for its reduction (removal). From the point that it is extremely difficult to validate what the noise is, but, correspondingly, what are the real data, the eye-view analysis is not suggested, even for experienced metrologists [10]. In the current state of knowledge, there are many even more sophisticated algorithms and procedures for surface roughness evaluation. The existence of all of them is entirely justified, nevertheless, the application of each of them requires mindful users, and even if a precise measurement technique (measuring device) is used, the errors in the ISO 25178 parameters can arise when each of the approaches is implemented erroneously. Regarding this, the precise measurement method also requires an accurate data processing procedure.

It was previously presented that in the high-frequency measurement error detection (definition of S-surface) process are valuable methods available in the commercial software such as PSD, ACF or TD approaches [10,17,46,51]. When defining it to the NS [34], these approaches can be suitable for both areal (3D) [58] or profile (2D) [29] roughness analysis.

When detecting high-frequency measurement errors, it has been observed that the shape of the centre part of the ACF is different, if the noise occurs [46]. In practice, the maximum (centre) value of ACF increases more rapidly when the high-frequency noise is found against when those errors are not defined.

Currently, it has been observed that if the received raw measured data were filtered with a cutoff equal to the three values of the sampling or spacing (depending on the stylus or optical measuring method) and if the NS contained other than noise features (e.g., scratches or valleys), the amplitude of the high-frequency measurement error is too small to significantly influence the values of the ISO 25178 parameters. For this type of investigation, the FFTF was applied. Its suitability for high-frequency noise suppression has already improved [29]. From this suggestion, if the sampling interval is 5 μm (stylus method) and the spacing is 3.27 μm (optical measurement), the cutoff value should not be smaller than 15 μm and 10 μm, respectively. This dependence was already proposed in defining the S-nesting index, which should be set at a 3:1 ratio with the maximum sampling distance [91]. However, the nesting index can be clearly beneficial in many metrological issues [92–95], where the L- or S-filtering characteristics were applied.

According to the NS properties, first, it was observed that it only contained the high-frequency components. This characteristic can be evaluated by the use of a PSD function, nevertheless, the thresholding methods can be a suitable alternative [46,64]. In fact, proper NS received by the removal of high-frequency measurement errors from the results of the surface topography measurements should only include features in the high-frequency domain. Therefore, the PSD graph should only contain the features with the smallest value, which are indicated as those in the required (high-) frequency domain.

Second, the NS in the high-frequency domain, if it does not contain features with other frequencies than in the high-frequency domain, should be isotropic, despite the directionality of the surface topography studied. The occurrence of the non-noise [51,64] features can considerably reduce the isotropic properties of the surface. For the inspection of the directionality of the NS, the TD graph can be extensively used. In Figure 7, the NS received from filtration of the turned cylindrical surface with additionally burnished deep and wide dimples is presented. Various available commercial software filters were applied. From the previous property, when PSDs were considered, it was found that NSs created by SF (Figure 7h) and FFTF (Figure 7n) provided the most encouraging results. However, when considering the isotropic requirements, the application of the FFTF scheme (Figure 7o) seems to be a better solution for the suppression of high-frequency measurement errors from the results of the surface topography measurements. For the precise definition of the S-surface, we suggest analysing the PSD, ACF, and TD of the N, received after a digital filtration.

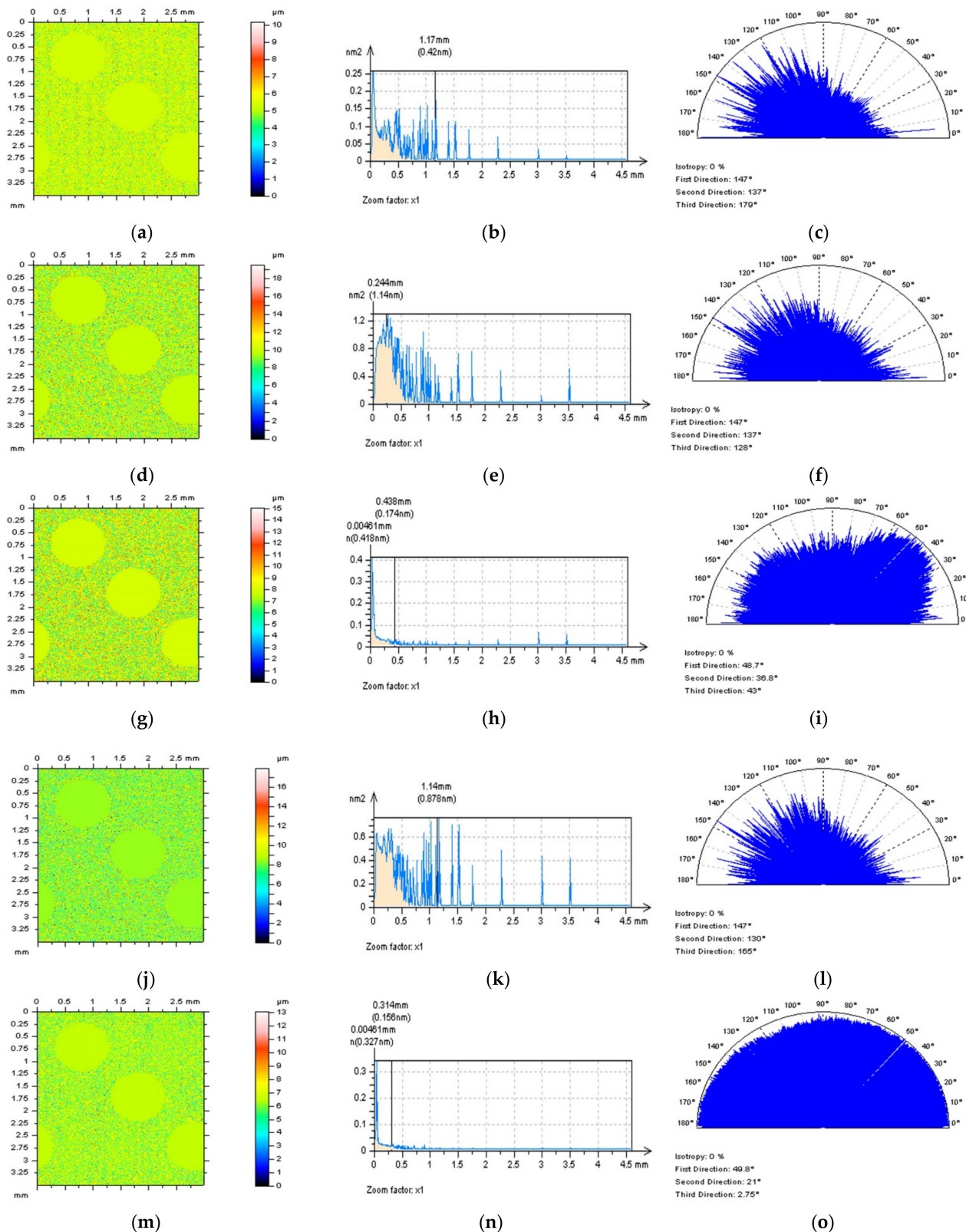

**Figure 7.** Isometric views of the S-surfaces (left column), their PSDs (middle), and TD graphs (right column) received after the application of GRF (**a**–**c**), RGRF (**d**–**f**), SF (**g**–**i**), MDF (**j**–**l**), and FFTF (**m**–**o**) method with a cutoff = 0.025 mm.

### 3.3. Proposals of Procedures for a Precise Definition of the S-L Surface with a Validation Concerning the (2D) Profile Data

The validation of procedures for the L-surface and S-surface definition can also be studied if dependent on the sequence of the algorithm provided. It was found in previous studies [17] that the occurrence of high-frequency measurement noise did not affect the validity of the method for areal form removal, resulting in defining the L-surface. Regarding this, we suggest first removing the form and then applying a filter for high-frequency measurement noise suppression. When defining the S-L surface, the L-surface can be defined in advance of the S-surface. Moreover, many proposals (procedures) for the definition (detection) and removal (reduction) of high-frequency measurement errors can be more validated when a flat (not containing form) surface is considered. Therefore, the procedure of the S-surface definition can be more valuable when applied after the definition of the L-surface.

The definition of both the L-surface and S-surface can be even more valuable when considering analysing the 2D (profile) data. Profile characterisation can be crucially important when the distortion of the valleys is considered. It was previously found that exaggeration in areal form removal is even easier to detect when dimples are located near/on the edge of the analysed data [43]. However, the distortion can also be visible for each of the surfaces containing deep and wide oil pockets [16,40]. Moreover, the density and distribution can affect the suitability of the L-surface method [17]. Surfaces containing many dimples, despite their sizes, can find it difficult to define the L-surface by excluding the valleys, such as the VEM scheme [43]. In this case, the process of feature exclusion may be too time-consuming. Moreover, the excluded area may be too large for the area that is not. One of the solutions is to enlarge the value of the cutoff (e.g., for the 2.5 mm or larger [96–98]). The flatness of the dimples should be reduced, nevertheless, in form, especially those components that reflect the waviness, which would not be extracted.

In Figure 8, various circumferential profiles received from the plateau-honed cylinder liner surface are presented. They differed by the number of dimples and valleys included, from a profile containing three dimples and two valleys (Figure 8a) to a profile with one dimple and one valley (Figure 8f). Characterising and definition of the S-L surface are usually highly demanding for surfaces containing the largest number (and density, respectively) of the features and dimples in recent studies. Regarding this, the profile containing five features was analysed.

For surfaces and profiles with a huge number of dimples, usually problems in the precise definition of the S-L surface derive from the errors in the areal form removal and L-surface calculation. In Figure 9a, the parts of the profile containing features, dimples, or valleys (scratches) are indicated by the arrows. It can be observed that, relatively, this part of the profile was around 50% of the whole profile length. In this case, the possibility of errors in positioning the reference line can increase enormously. Each of the Gaussian (GRF or RGRF) or spline (SF) methods caused serious distortions of these features. Deviation from the required results can be observed with the deviations in the reference line (Figure 9f–j)—this was far from the cylindrical. More relevant results were obtained when the least-square fitting methods (POLY2 and POLY4) were applied. From the analysis of both profiles (Figure 9a,c), the one received by the usage of POLY2 seemed to be flatter and, respectively, the total height of the profile (*Pt*) was smaller. However, for surfaces containing oil pockets, the minimisation of the maximum height is not always required, and correspondingly, in some cases, can indicate the flatness of some features, especially deep and wide dimples [14,16,40]. This can affect the classification of properly made parts as a lack.

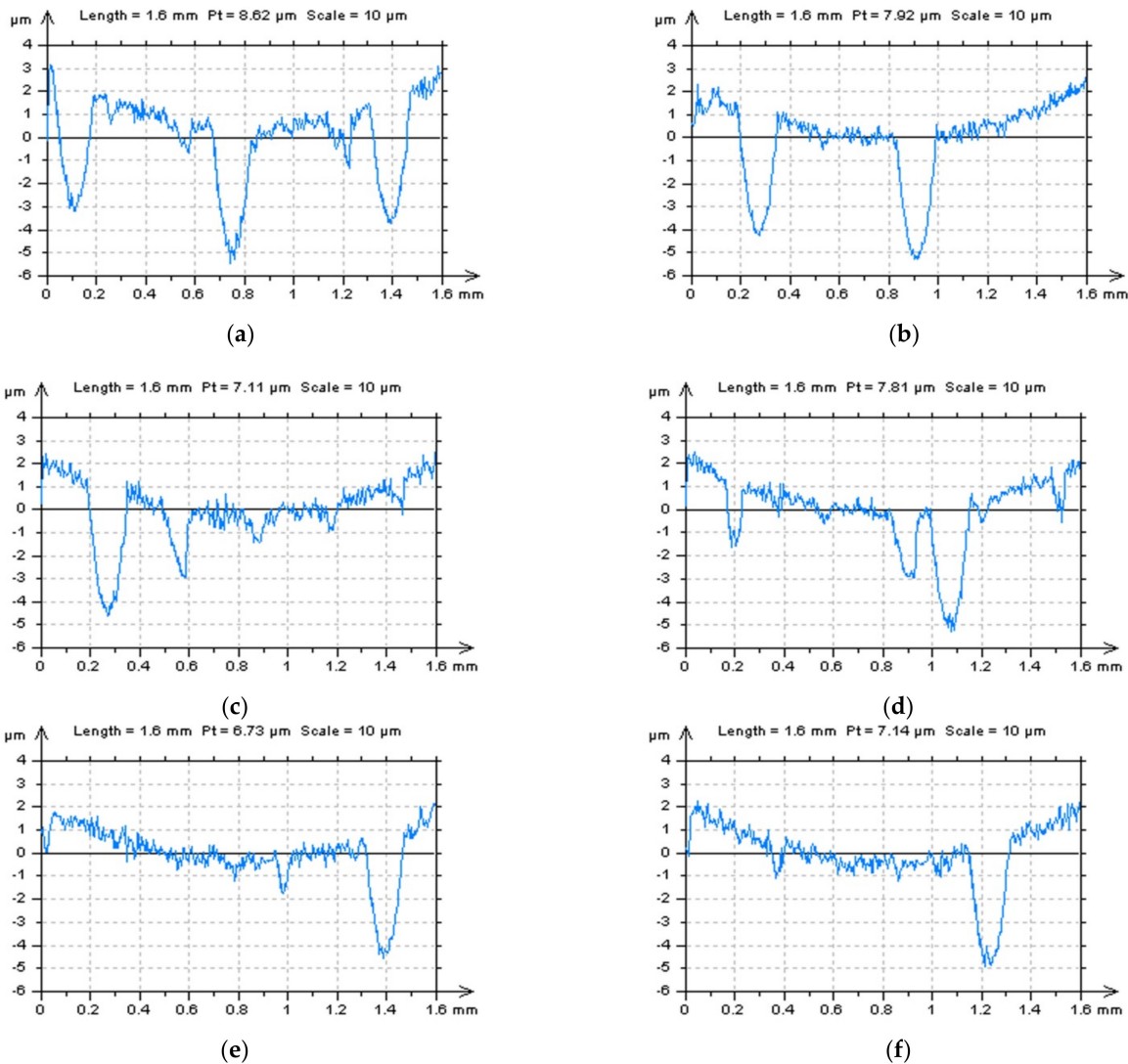

**Figure 8.** Examples of profiles extracted from the raw measured data of the plateau-honed cylinder liner surface topography including three dimples and two valleys (**a**), two dimples and one valley (**b**), one dimple and four valleys (**c**), one dimple and three valleys (**d**), one dimple and two valleys (**e**), and one dimple and one valley (**f**).

Profile of the surface after L-surface definition by the POLY2 (Figure 9a) method was further considered for high-frequency measurement noise detection and reduction, and the definition of the S-surface. Results after various filtering (GRF, RGRF, SF, MDF and FFTF) were compared, analysed, and are presented in Figure 10. In the presented example, differences were observed for each of the data presented, the S-L profile (left column), S-profile (middle), and PSD graphs, defined for the S-profiles. From the studies of S-profiles received by the application of GRF (Figure 10b), RGRF (Figure 10e), and SF (Figure 10h), some extraordinary features were found, examples of which are indicated by the arrows. These features, located in other than a high-frequency domain, were visible on the PSD graphs. The S-profile defined by the MDF application contained, comparatively, areas where the noise did not occur, also indicated by the arrow. This seems to be out of the logic that, respectively, the vibration, which is usually affecting the high-frequency measurement noise occurrence, influences the whole measurement data, and not only those selected. However, it was not considered in this or previous studies. Out of all of the filters proposed for the reduction in high-frequency measurement errors, the application of the FFTF seems to be the most suitable.

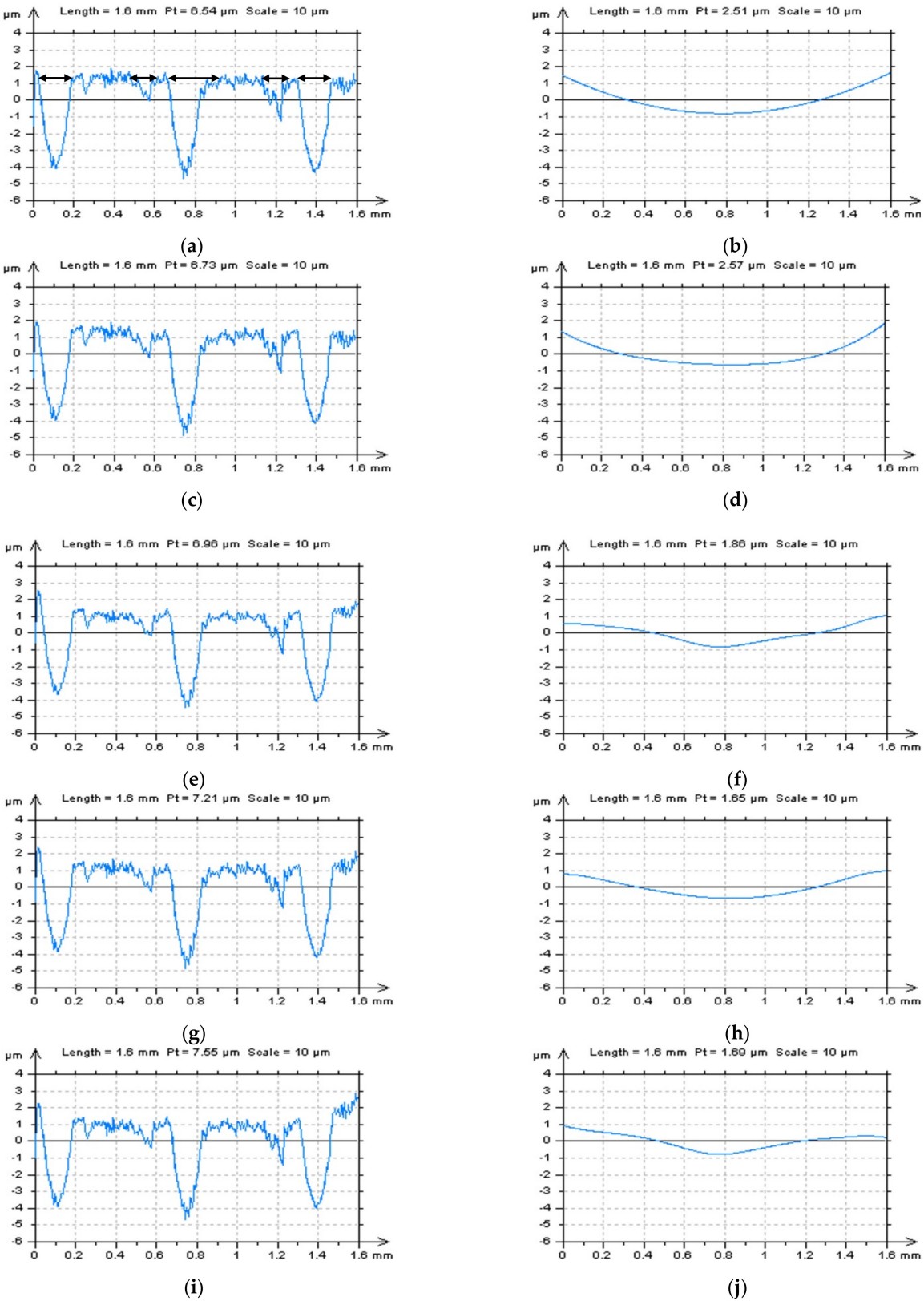

**Figure 9.** Profiles after L-surface definition (left column) and form removal (right column) received from the example presented previously in Figure 8a by application of Poly2 (**a,b**), Poly4 (**c,d**), GRF (**e,f**), RGRF (**g,h**), and SF (**i,j**), cutoff = 0.8 mm.

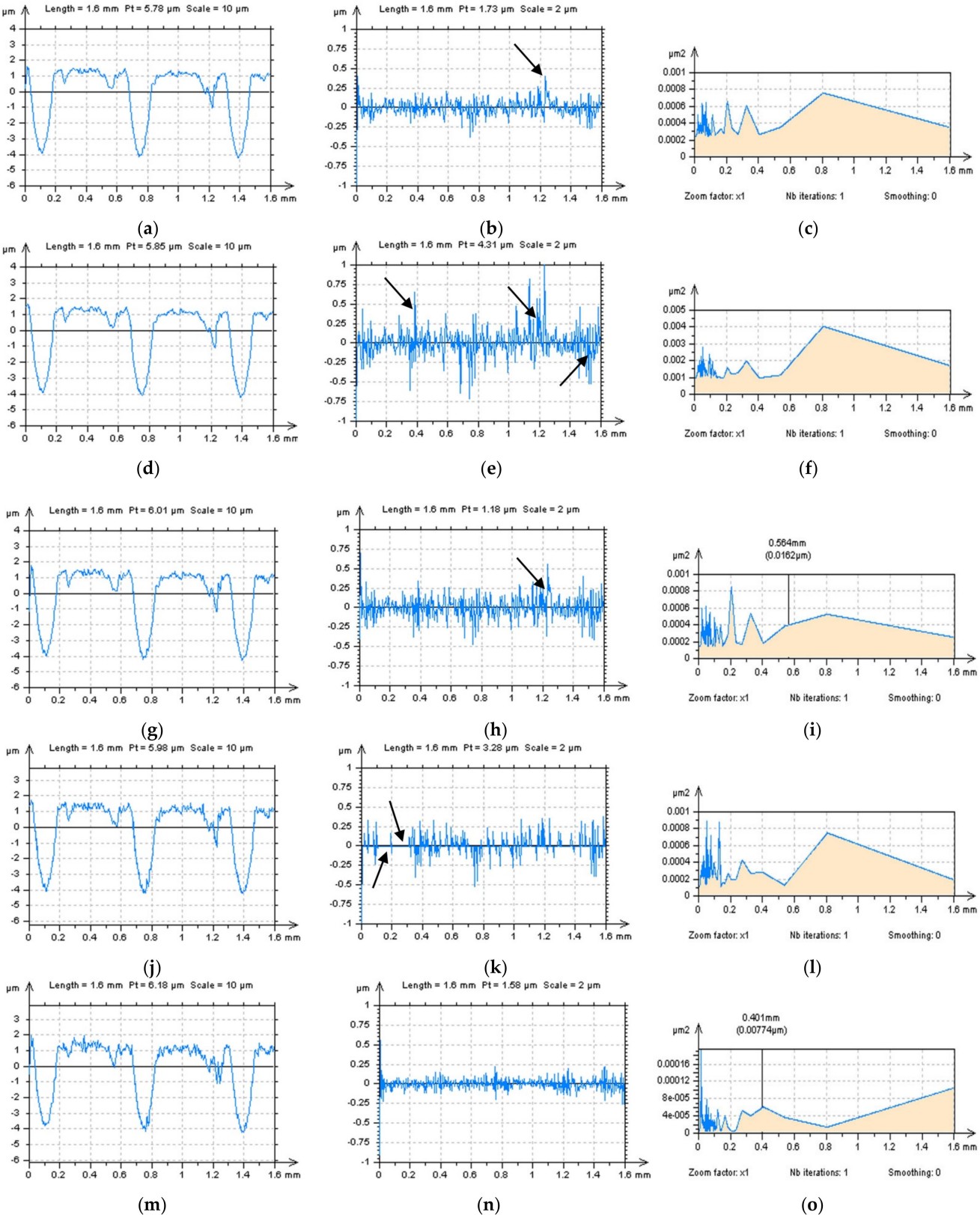

**Figure 10.** Profiles after S-L surface definition (left column), removed S-surface (middle), and its PSD (right column) after the application of GRF (**a–c**), RGRF (**d–f**), SF (**g–i**), MDF (**j–l**), and FFTF (**m–o**) method with a cutoff = 0.015 mm.

From the above analysis, for the removal of form and a reduction in the high-frequency measurement noise, derived from the plateau-honed cylinder liner surface containing oil pockets (dimples) created by the burnishing techniques, the most encouraging methods (from all of those considered) were POLY (for L-surface definition) and FFTF (S-surface calculation), when the S-L surface needs to be precisely determined.

## 4. The Outlook

Despite the many proposals suggested, there are still many issues that should be considered in future studies, as follows:

1.  Areal form removal of other cylindrical surfaces, excluding plateau-honed, turned, or isotropic, should also be carefully considered. In the presented studies, the most important was an assessment of the dimples and their distortions (if occurred) when selecting both the L-surface and S-surface.
2.  Detection of high-frequency measurement noise in the process of S-surface definition should be proposed with a method excluding deep and wide features such as analysed dimples. The precision in the thresholding of the oil pockets should also be considered. Some examples of these studies were considered previously, nevertheless, they must be further and more comprehensively analysed.
3.  Reduction of errors in the roughness evaluation considering a precise definition of the S-L surface should be proposed for other types of surfaces (e.g., ground, milled, laser-textured, composite, ceramic, or many others). Problems in the definition of the S-L surface can be different for each type of analysed surface.
4.  Validation of general, commercially available software, methods, and procedures should be improved more significantly. Currently, there are many, even more, sophisticated approaches that make it extremely difficult to propose one general procedure for a precise roughness evaluation.

## 5. Conclusions

According to the whole analysis presented, the following comments can be defined:

1.  Analysis of the surface topography and calculation of the ISO 25178 roughness parameters are dependent on the precision in the definition of the S-L surface. The whole process of S-L surface selection can be roughly divided into proposals of L-surface and S-surface.
2.  All of the commonly used (available in commercial software) methods were found to be suitable for the definition of the S-L surface such as least-square fitted cylinder or polynomial plane of nth degrees, regular and robust Gaussian regression filters, regular isotropic spline filter, and fast Fourier transform filter. Nevertheless, the most encouraging issue is to apply them appropriately. Improvements in their application were found with support by autocorrelation function, power spectral density, and texture direction graph.
3.  From the analysis of the cylindrical surfaces with additionally burnished dimples, it was found that the order of the surface definition (first the L-surface and then the S-surface) did not affect the precision in the surface roughness parameter calculation. However, for the definition of high-frequency measurement noise, the flat (after an areal form removal process) surface was more relevant for measurement error detection. Therefore, we first suggest selecting the L-surface, and then the S-surface.
4.  When considering surfaces containing burnished features, like oil pockets, and dimples, the selection of an L-surface can be a demanding task that the errors in roughness parameters can be enlarged. Usually, this is caused by a distortion of the dimples. It was observed that deep and wide features can radically affect the position of the reference plane (line) for the areal (profile) data.
5.  For surfaces containing deep and wide features such as dimples, it was found that digital filtering (e.g., various Gaussian (GRF and RGRF) or spline (SF) filters) can cause serious distortion of the values of the surface roughness parameters. The

distortion of valleys was especially visible when considering the profile data, nevertheless, areal surface topography parameters (like *Sp, Sv, Spk, Svk* and *Sk*) were also falsely estimated.

6. Detection and reduction of high-frequency measurement errors, when defining the S-surface, can be fraught with many errors related to the occurrence of dimples. We suggest selecting areas (or profile parts in 2D considerations) where oil pockets are not located.

7. Reduction in the high-frequency measurement noise was proposed with an application of various functions such as ACF, PSD, and TD. These techniques are essential in the characterisation of the noise surface, represented by the S-surface.

**Funding:** This research received no external funding.

**Institutional Review Board Statement:** Not applicable.

**Informed Consent Statement:** Not applicable.

**Data Availability Statement:** Data sharing is not applicable to this article.

**Conflicts of Interest:** The author declares no conflict of interest.

## Nomenclature

The following abbreviations and surface topography parameters are used in the manuscript:

| | |
|---|---|
| *ACF* | Autocorrelation function |
| *AFM* | Atomic force microscopy |
| *FFT* | Fast Fourier transform |
| *FFTF* | Fast Fourier transform filter |
| *FIR* | Finite impulse response filter |
| *GRF* | Regular Gaussian regression filter |
| *L-filter* | Filter used for definition of the L-surface |
| *L-surface* | Surface received after L-filtering |
| *LSFC* | Least-square fitted cylinder reference plane |
| *MDF* | Median denoising filter |
| *NS* | Noise surface |
| *POLY2* | Reference plane obtained by the least-square fitted polynomial of the second degree |
| *POLY4* | Reference plane obtained by the least-square fitted polynomial of the fourth degree |
| *POLY6* | Reference plane obtained by the least-square fitted polynomial of the sixth degree |
| *PSD* | Power spectral density |
| *RGRF* | Robust Gaussian regression filter |
| *RMS* | Root mean square (roughness) |
| *S-filter* | Filter used for definition of the S-surface |
| *S-L surface* | Scale-limited surface received after S- and L- filtering |
| *S-surface* | Surface received after S-filtering |
| *SF* | Regular isotropic spline filter |
| *SSA* | Singular spectrum analysis |
| *TD* | Texture direction (graph) |
| *VEM* | Valley excluding method |
| *Sa* | Arithmetic mean height Sa, μm |
| *Sal* | Auto-correlation length, mm |
| *Sbi* | Surface bearing index |
| *Sci* | Core fluid retention index |
| *Sdq* | Root mean square gradient |
| *Sdr* | Developed interfacial areal ratio, % |
| *Sk* | Core roughness depth, μm |
| *Sku* | Kurtosis |
| *Smc* | Inverse areal material ratio |
| *Smr* | Areal material ratio |

| | |
|---|---|
| *Sp* | Maximum peak height, μm |
| *Spc* | Arithmetic mean peak curvature, 1/mm |
| *Spd* | Peak density, $1/mm^2$ |
| *Spk* | Reduced summit height, μm |
| *Sq* | Root mean square height, μm |
| *Ssk* | Skewness |
| *Std* | Texture direction, ° |
| *Str* | Texture parameter |
| *Sv* | Maximum valley depth, μm |
| *Svi* | Valley fluid retention index |
| *Svk* | Reduced valley depth, μm |
| *Sxp* | Extreme peak height |
| *Sz* | The maximum height of surface, μm |

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
