# Peer review of "Roughness Evaluation of Burnished Topography with a Precise Definition of the S-L Surface"

_applsci, doi:10.3390/app122412788_

Round 1
Reviewer 1 Report
The paper aims to provide guidance on how to provide accurate S-L surface roughness parameters and false estimation of parameters. Overall, the article is interesting, however, sometimes is hard to read. Some suggestions are proposed to improve its quality and text flow.
1) Introduction
Line 32 – 34
Line 40
Line 41 – 42
Line 68 - 71
These statements are not clear. There are spelling and grammar errors. Please revise.
The last paragraph should clearly identify the research gap which this paper will address.
Materials and Methods:
Line 140
Is there any particular reason why 20% was selected?
Line 169 – reference missing
Section 2.3 is quite difficult to read. Many methods are introduced, and the text jump from one to another. Can the author include a Table or a schematic to highlight which methods were used?
Line 245
Can the author elaborate on why?
Results
Line 400 - spelling
Conclusion:
Please revise statement 1, spelling.
Author Response
Dear reviewer,
Please find in tha attached file, I hope, all of the suitable responses to the comments raised.
Best regards,
Author

Reviewer 2 Report
This paper has some reference value for readers. The revision suggestions are as follows:
1. S-L is the abbreviation form,should give the full name first,then give the abbreviation form, including the title, abstract and manuscript. Please chect the other abbreviation words for the whole paper.
2. Previous studies were referred severial times in the paper, and the references should be given.
3. The middle part of Fig. 5 is missing part and needs to be completed.
4. The description of the applied methods can be simpler.
Author Response

(The authors gave the same response as above.)
